# Development of Cereal Bars Enriched with Andean Grains and Patagonian Calafate (*Berberis microphylla*): Nutritional Composition, Phenolic Content, Antioxidant, Textural, and Sensory Evaluation

**DOI:** 10.3390/foods14234127

**Published:** 2025-12-02

**Authors:** Jéssica López, Romina Cea, Nicole Tiznado, Evelyn Fernández, María Lorena González, Sebastián Pizarro-Oteíza, Carmen Pérez-Cervera

**Affiliations:** 1School of Food Science, Pontificia Universidad Católica de Valparaíso, Waddington 716, Playa Ancha, Valparaíso 2360100, Chile; romicear72@gmail.com (R.C.); evelyn.fernandez.a@mail.pucv.cl (E.F.); maria.gonzalez.r@pucv.cl (M.L.G.); sebastian.pizarro@pucv.cl (S.P.-O.); 2Department of Health, Community and Management, Faculty of Health Sciences, Universidad de Playa Ancha, Valparaíso 2340000, Chile; nicoletiznadocastillo.nut@gmail.com; 3Agro-Industrial Engineering Program, Pontificia Bolivariana University, Montería 230002, Colombia; carmenelena.perez@upb.edu.co

**Keywords:** cereal bar, *Amaranthus* sp., antioxidant, *Berberis microphylla*, *Chenopodium quinoa*, nutritive value

## Abstract

**Background**: Cereal bars are convenient vehicles for incorporating ingredients with functional value. In this context, the study aimed to formulate bars enriched with quinoa, amaranth, and calafate (*Berberis microphylla*) and evaluate their instrumental texture, total phenolic compounds, antioxidant capacity, nutritional composition, and sensory evaluation. **Methods**: Four formulations were developed, a baseline cereal bar with balanced ingredients (F1), a pseudocereal-enriched bar (F2), a high-calafate bar (F3), and an oat-only control bar (F4). Texture was measured using uniaxial compression, total phenolic compounds (TPC) were determined by the Folin–Ciocalteu method, and antioxidant capacity was assessed by the DPPH assay. The nutritional composition was theoretically estimated using food composition tables and dietary reference intakes (DRIs). Sensory evaluation was performed using affective tests, including acceptability, preference, purchase intention, and sensory attributes. **Results**: The formulations differed significantly in instrumental hardness. F3 had the highest total phenol content and the highest antioxidant capacity. The estimated nutritional composition showed that the bars provide adequate energy and relevant micronutrients (Ca, Mg, Fe, Zn), as well as bioactive compounds from calafate. Sensory evaluation showed that F2 obtained the highest overall acceptance and the highest acceptability index. Purchase intention did not differ between formulations. In the evaluation of attributes, the results indicate that intermediate hardness maximizes acceptance, while softer (F1) or harder (F3–F4) bars are less preferred. **Conclusions**: The incorporation of calafate enhances the phenolic and antioxidant profile in F3, while an intermediate hardness linked to the greater use of expanded pseudocereals favors consumer acceptance in F2. The observed differences confirm that the formulation design enables the modulation of functional, mechanical, and sensory properties in cereal bars.

## 1. Introduction

In recent years, there has been growing interest in developing functional foods that combine traditional ingredients with bioactive and nutritional properties. Snacks in the form of cereal bars are a practical option that is highly valued by consumers looking for healthy, nutritious, and easy-to-eat foods. The versatility of their design allows for the incorporation of local raw materials, which promotes the revaluation of Latin America’s agricultural and cultural heritage [1,2].

Andean grains, such as quinoa (*Chenopodium quinoa*) and amaranth (*Amaranthus* spp.), are notable primarily for their nutrients, such as proteins, essential amino acids, dietary fiber, fatty acids, B vitamins, calcium, magnesium, zinc, and iron. In addition, they contain bioactive compounds such as flavonoids and polyphenols, which give them antioxidant and anti-inflammatory properties [3,4]. The thermal expansion or “popping” of these grains improves their texture while essentially preserving their nutritional and functional value. These qualities make them ideal ingredients for the design of foods that contribute to the prevention of chronic noncommunicable diseases [4,5]. In this context, the high dietary fiber content of these grains deserves particular attention, given that higher fiber intake has been associated with lower all-cause, cardiovascular, and cancer mortality [6,7].

On the other hand, calafate (*Berberis microphylla*), a fruit native to Chilean Patagonia, is distinguished by its intense pigmentation and high content of phenolic compounds, including anthocyanins such as delphinidin and cyanidin. These molecules have demonstrated antioxidant and anti-inflammatory effects and potential cardiovascular benefits [8,9]. The incorporation of calafate berries into food matrices is a strategy for adding value to a wild fruit with a native identity. This contribution not only enriches the nutritional profile of the product but also enhances its sensory characteristics, generating a synergistic interaction with the other ingredients present in the cereal bar [1].

In this context, it is essential to evaluate how the incorporation of these functional ingredients, particularly Andean grains and calafate, is reflected in the sensory properties of the product, determining its acceptance and preference by consumers, since sensory analysis is a fundamental component in the comprehensive characterization of functional cereal bars, as it allows for the evaluation of color, aroma, flavor, texture and overall acceptability, factors that directly influence consumers’ purchase intention and loyalty [10]. Several studies have shown that the inclusion of grains such as puffed quinoa and amaranth not only improves the nutritional profile but also provides desirable textural characteristics such as crunchiness, lightness, and low bulk density [11]. These ingredients, combined with natural binders such as honey, allow the cohesion and hardness of the final product to be adjusted. For this reason, the use of instrumental texture evaluation, through analysis of breaking force and hardness, is key to ensuring the sensory acceptability of the product [2,12].

In this regard, sensory perception is strongly influenced by the presence of characteristic fruit components. Ingredients such as calafate not only provide an intense and attractive color due to their high concentration of anthocyanins, but also a characteristic slightly acidic flavor that balances the sweetness of the ingredients in the cereal bar as a final product. This aspect is crucial in formulations aimed at adult consumers, athletes, and consumers who choose to lead a healthy lifestyle and are looking for foods with healthy nutritional attributes and specific sensory profiles [9,13]. In addition, sensory studies that use 9-point hedonic scales, such as FACT, provide detailed information on acceptability and possible improvements in formulation [14]. Thus, bars made with local ingredients tend to be viewed positively in contexts where the regional origin and naturalness of the ingredients are emphasized, key factors in current conscious consumption trends. The presence of bioactive compounds such as phenolic compounds and fiber, as well as the absence of artificial additives, reinforces the perception of health and well-being, which translates into a higher level of acceptance [15]. Therefore, sensory analysis not only allows for the optimization of the product profile but also for the adaptation of development strategies to the requirements and preferences of the target audience [16].

Consequently, this study proposes the development of cereal bars formulated with oats, Andean pseudocereals (quinoa and amaranth), and Patagonian calafate, with the aim of enhancing the value of these regional crops as functional ingredients. To our knowledge, there are no previous studies that simultaneously evaluate the nutritional composition, phenolic content, antioxidant capacity, texture, and consumer acceptance of cereal bars enriched with calafate. In this context, the objective of this study was to formulate a cereal bar using Andean grains (amaranth and quinoa) and native berries (specifically *Berberis microphylla*, commonly known as calafate), in order to analyze its instrumental texture, total polyphenol content, antioxidant capacity, nutritional composition, and to evaluate its acceptance by potential consumers through sensory evaluation.

## 2. Materials and Methods

### 2.1. Raw Materials and Formulation of Cereal Bars

The raw materials used to develop cereal bars were rolled oats, dried cranberries (*Vaccinium macrocarpon*), amaranth (*Amaranthus* sp.) popcorn, quinoa (*Chenopodium quinoa*) popcorn, honey, sunflower oil, and powdered calafate (*Berberis microphylla*) purchased from a local supermarket. All ingredients were used as received, without additional pretreatment in the laboratory. The composition of the different formulations is shown in Table 1.

Four different formulations of cereal bars were developed with the aim of evaluating the influence of different ingredients on the nutritional and bioactive properties of the final product. The formulations were designed by keeping certain components constant (cranberries, honey, and sunflower oil) while varying the proportions of oats, pseudocereals (amaranth and quinoa), and calafate.

The variation in pseudocereals and berries was based on their superior nutritional value. Amaranth and quinoa were incorporated for their high biological protein quality and balanced profile of essential amino acids, while calafate was selected as a native Chilean berry for its significant content of bioactive secondary metabolites, including antioxidants, polyphenols, and flavonoids, constituting a natural source of functional compounds.

For each formulation, cereal bars were produced in three independent batches. The cereal bar production process was carried out by mixing the ingredients in two stages: first, the dry ingredients (oats, expanded amaranth, expanded quinoa, dehydrated cranberries, and calafate) were combined and mixed homogeneously for 3 min; then the wet ingredients (honey and sunflower oil) were added, and the mixture was continued for an additional 5 min using a mixer until a homogeneous and cohesive dough was obtained. The prepared dough was distributed evenly in silicone molds previously greased with sunflower oil, measuring 11 × 4 × 2 cm, to ensure the uniformity of the final product. The bars were baked in a convection oven at 180 °C for 15 min. After baking, the molds with the bars were cooled at room temperature (20 ± 2 °C) for 15 min. Finally, the bars were removed from the molds, were packed in polyethylene-coated aluminum foil, and stored at room temperature for subsequent analysis.

Four different cereal bar formulations were developed according to the proportions detailed in Table 1. The visual appearance of the four cereal bar formulations is presented in Figure 1.

### 2.2. Determination of Texture

The texture analysis of the cereal bars was performed using a TA.XT2 texture analyzer (Stable MicroSystems, Godalming, UK) equipped with a 25 mm diameter stainless steel cylindrical probe. The texture parameters were determined using the XTRAD software Texture Expert, version 7.16 (Stable Micro Systems, Godalming, UK) integrated into the equipment. For texture analysis, rectangular pieces of about 4 × 4 × 2 cm were cut from the central region of each bar. The operational conditions of the test were established as follows: pre-test speed of 1 mm/s, test speed of 0.5 mm/s, post-test speed of 1 mm/s, with a penetration distance of 10 mm. The analysis was performed by uniaxial compression with controlled deformation, recording the force-displacement curve during the process. The primary textural outcome was hardness (peak force, N), calculated as the highest force reached during the 10 mm compression. For each formulation, a minimum of 10 repetitions were analyzed, selecting representative portions from the center of each bar to avoid edge effects.

### 2.3. Determination of Bioactive Compounds of the Cereal Bars

#### 2.3.1. Preparation of Cereal Bars Extracts

The procedure developed by Silva and Conti-Silva [17] was used for the extraction, with slight modifications. In summary, 2 g of ground samples were mixed with 40 mL of ethanol/water solution (50/50, *v*/*v*) and extraction was carried out in a shaker at room temperature (22 ± 2 °C) for 60 min. After centrifuging at 25,400 g (Thermo IEC HNS-II, Needham Heights, MA, USA) for 15 min, the supernatant was removed, and the extraction was repeated once more in a similar manner for 60 min in a shaker. The supernatants obtained from both extractions were combined and filtered through filter paper using a #1 filter (Whatman International Limited, Maidstone, Kent, UK) and an analytical funnel. The filtered extract was brought to a final volume of 100 mL in a volumetric flask using the same extraction solvent. All extractions were performed in triplicate and were then used to determine the quantity of total phenolic compounds and the antioxidant capacity.

#### 2.3.2. Determination of Total Phenolic Compounds (TPC)

Total phenolic compounds were determined by spectrophotometry using the Folin–Ciocalteu method, based on the colorimetric oxidation-reduction reaction, according to the protocol of Singleton and Rossi [18] with modifications. 5 μL of extract was transferred to the wells of a 96-well microplate, followed by the addition of 75 μL of distilled water and 25 μL of 1 N Folin–Ciocalteu reagent. After 3 min, 30 μL of Na_2_CO_3_ solution (10% *w*/*v*) was added, followed by 120 μL of distilled water. The samples were incubated for 2 h in the dark at room temperature before measuring the absorbance at 760 nm in a microplate reader (Thermo Fisher Scientific Multiskan GO, Vantaa, Finland) and compared to a gallic acid equivalent (GAE) calibration curve. The results were expressed as mg of GAE/100 g of sample. All determinations were performed in triplicate, and the data were reported as mean ± standard deviation.

#### 2.3.3. Determination of Antioxidant Capacity Measured Using DPPH Radical Scavenging Activity

Antioxidant capacity was determined using the DPPH (2,2-diphenyl-1-picrylhydrazyl) free radical scavenging assay, following the method of Moreira et al. [19] with modifications. 100 μL of the extract was mixed with 100 μL of DPPH solution (154 μM prepared in ethanol). The reaction mixture was shaken vigorously for a few seconds and incubated at room temperature in the dark for 30 min. After the reaction time had elapsed, the absorbance was measured at 517 nm in a microplate reader (Thermo Fisher Scientific Multiskan GO, Vantaa, Finland) and compared to a Trolox equivalent (TE) (6-hydroxy-2,5,7,8-tetramethylchroman-2-carboxylic acid) calibration curve. The results were expressed as μM of TE/100 g of sample. All determinations were performed in triplicate, and the data were reported as mean ± standard deviation.

### 2.4. Theoretical Estimation of Nutritional Composition

For the development of the nutritional labeling table, the calculation of the contribution of macronutrients (energy, protein, fats, and carbohydrates) and micronutrients (vitamins and minerals) per serving was initially carried out, using the chemical-nutritional composition information available in the Food Composition Table of the Institute of Nutrition and Food Technology (INTA) [20], and nutritional tables based on USDA data [21]; for micronutrients, reference values were obtained from the Dietary Reference Intakes (DRIs) defined by National Academy of Medicine [22], considering the profile of a healthy average adult. Subsequently, the percentage contribution of each macronutrient was calculated based on a standard 2000 kcal diet, and for micronutrients, the percentage of daily requirement (DRI) met by a serving of the evaluated food was estimated.

### 2.5. Sensory Analysis

The sensory evaluation of the cereal bar samples was carried out using affective tests with a panel of 35 untrained adult consumers (aged 18–55). All procedures were developed according to the Guide for Sensory Analysis ISO 6658:2017 [23], and the consumers participating in the evaluation received Information for Study Participants and completed an Informed Consent Form for Participation in the Study. The evaluations were carried out in a standardized sensory laboratory with white LED lighting and controlled temperature (20 ± 2 °C). Mineral water at room temperature was provided as a neutralizing agent between sample tastings. The samples were presented in a random and balanced manner, coded with random three-digit numbers to avoid order and identification bias.

#### Applied Affective Tests

Acceptance: The 9-point Food Action (FACT) rating scale was used to evaluate product acceptance, with options ranging from “I would only eat this if I had to” (1 point) to “I would eat this whenever I had the chance” (9 points). This scale allows product acceptance to be evaluated based on the frequency of consumption that the panelist would be willing to have. In addition, the acceptability index (AI) was calculated using Equation 1:
(1)$$AI(\%)=\frac{A}{B}\times 100$$ where A represents the average score given for the product, and B the maximum score obtained for the product. AI of 70% is considered indicative of good product acceptance [24].

Preference: The preference test was conducted using a 9-point verbal hedonic scale (“extremely dislike” (1) to “extremely like” (9)) to assess overall product acceptance.

Intention to purchase: A 5-point hedonic scale was used to assess purchase intention (“definitely would not buy” (1) to “definitely would buy” (5)).

The samples were delivered coded with four-digit numbers, and a 9-point hedonic scale (1 = dislike extremely, 9 = like extremely) was used to carry out the acceptance test (31) for aroma, color, texture, flavor, and overall acceptability.

Sensory attributes: Verbal hedonic scales were used to evaluate appearance, taste, texture, aroma, and color attributes, using 5-point scales (1 = dislike extremely, 5 = like extremely).

### 2.6. Statistical Analysis

Statistical analysis was performed using software Statgraphics^®^ Centurion 18 (The Plains, VA, USA) using analysis of variance (one-way ANOVA). The significant differences (*p* < 0.05) between the means were performed using the least significant difference (LSD) test at a significance level of α = 0.05 and a confidence interval of 95% (*p* < 0.05). A multiple range test (MRT) was also performed to demonstrate the existence of homogeneous groups within each of the parameters.

## 3. Results and Discussion

### 3.1. Texture Determination

The texture of the cereal bar formulations was determined by the maximum compression force (peak force; hardness, N), with the results presented in Table 2. Significant differences (*p* < 0.05) were observed between the formulations. The maximum force required for compression varied considerably between formulations, ranging from 7.31 ± 1.24 N for F1 to 27.63 ± 0.18 N for the control formulation F4. The control formulation (F4) exhibited the highest hardness, followed by F3 (20.27 ± 2.33 N), F2 (11.96 ± 2.04 N), and F1 (7.31 ± 1.24 N). These significant differences in maximum strength can be attributed to differences in the composition of the formulations. For example, the control formulation (F4), which contains the highest proportion of oats (240 g) without pseudocereals or calafate, showed the highest hardness. These results are consistent with the structural properties of oats, which contain β-glucans and other fibrous components that contribute to matrix cohesion and firmness during processing and cooling [25]. Formulation F3, which contained twice the amount of calafate (48 g) compared to F1 and F2 (24 g), showed intermediate hardness values. The increase in calafate content may have contributed to improving structural integrity through the interaction of its fibrous components and natural binding properties with the cereal matrix [26]. The presence of bioactive compounds in calafate, including polyphenols and dietary fiber, may have influenced intermolecular interactions within the bar structure, resulting in greater resistance to deformation. Similar interactions between phenolic compounds and polysaccharides have been demonstrated to enhance the structural integrity and modify the textural properties of cereal-based food systems [27,28].

Formulation F2, characterized by a reduced oat content (60 g) and an increase in the pseudocereal content (21 g each of amaranth and quinoa), exhibited moderate hardness values. The replacement of oats with expanded pseudocereals probably altered the overall density and cohesiveness of the matrix. Expanded pseudocereals typically have lower bulk density and different structural characteristics compared to rolled oats, which could explain the lower maximum force values observed [10,29]. In contrast, formulation F1, with balanced proportions of all ingredients, showed the lowest maximum strength values, suggesting a more tender texture.

The hardness values obtained in this study (7.31–27.63 N) are consistent with cereal bars reported in the scientific literature with similar textural characteristics. For example, McMahon et al. [30] reported hardness values of 15–24 N at manufacture (increasing during storage) in high-protein nutrition bars, Marques et al. [28], a study on acerola-enriched cereal bars, reported hardness of 26–56 N, and Iuliano et al. [31], a quinoa-bar study, obtained values of hardness of 24 N at day 0. This suggests that the formulations in this study have been developed to provide a cohesive yet easy-to-chew texture suitable for general consumption.

### 3.2. Total Phenolic Compounds (TPC) of Cereal Bars

The total phenolic compound (TPC) content was determined using the Folin–Ciocalteu method. As shown in Table 3, formulation F3 had the highest TPC value (403.49 ± 15.84 mg GAE/100 g), which was significantly greater than those of F1 (264.67 ± 9.76 mg GAE/100 g) and F2 (231.18 ± 10.63 mg GAE/100 g) (*p* < 0.05). In contrast, the control formulation F4 showed the lowest TPC (141.34 ± 9.73 mg GAE/100 g). This trend suggests that increasing the concentration of bioactive ingredients, such as calafate, enhances phenolic content.

These results can be attributed to differences in the composition of the formulations, as detailed in Table 1. The F3 formulation incorporates 48 g of calafate, twice the amount used in F1 and F2 (24 g each), which explains its significant increase in phenolic compound content. This result is particularly relevant, as calafate is a native Patagonian berry that has been little explored in cereal bars, and its incorporation into this matrix allows its functional contribution to be compared with that of conventional berries, since the proportion of cranberries remained constant in all formulations. Likewise, F3 has a balanced matrix composed of 120 g of oats and 14 g of quinoa and amaranth popcorn, which favours the concentration of bioactive compounds. In contrast, formulation F4, composed solely of 240 g of oats and without the addition of calafate or pseudocereals, has a low proportion of ingredients containing phenolic compounds, which is reflected in the lower values recorded.

The role of calafate is decisive, as this Patagonian berry is characterized by a high concentration of anthocyanins, flavonoids, and phenolic acids [9,32]. Several studies have shown that incorporating calafate into food matrices significantly increases the total phenolic content and antioxidant capacity, due to the synergy between anthocyanins and other bioactive metabolites [33,34].

The phenolic profile of calafate is notably distinctive among wild berries due to the presence of anthocyanidin 3,7-β-O-diglucosides, a structural form that contrasts with the more prevalent 3,5-diglucosides in other berries. The principal anthocyanins identified in calafate are delphinidin 3,7-diglucoside, petunidin 3,7-diglucoside, and malvidin 3,7-diglucoside, as well as their corresponding monoglucosides. This distinctive anthocyanin composition, together with substantial concentrations of hydroxycinnamic acid derivatives (such as caffeoyl-glucaric acids) and flavonols (mainly quercetin derivatives), underpins calafate’s remarkable antioxidant capacity. The glycosylated structure of these phenolic compounds improves their stability during food processing and storage, which is essential for functional food applications. Research indicates that calafate’s anthocyanin content (17.81 μmol/g fresh weight) is similar to that of maqui berry (*Aristotelia chilensis*), another Chilean superfruit, and significantly exceeds that of common berries, including blueberries, raspberries, and strawberries [32,35,36].

In addition, pseudocereals such as quinoa and amaranth contribute to a lesser extent but provide flavonoids and phenolic acids that reinforce the functional profile. Compared with quinoa as the main ingredient, the typical TPC observed in quinoa-based products ranges from 39 to 213 mg GAE/100 g [37], indicating that the phenolic content in F1 (264.7 mg GAE/100 g) and F2 (231.2 mg GAE/100 g) far surpasses that of quinoa alone. This increase points to the significant contribution of other bioactive ingredients, particularly calafate, which interacts synergistically to enhance the total phenolic content.

Formulation F3, which contains twice as much calafate compared to F1 and F2, had a significantly higher total phenolic compound content (403.49 ± 15.84 mg GAE/100 g), suggesting an additive effect attributable to this ingredient. Recent studies on functional bars enriched with red fruit powders or berry by-products have shown proportional increases in both total phenolic content and antioxidant capacity, with a positive dose–response relationship observed up to an acceptable sensory limit, beyond which organoleptic properties are compromised [38]. In particular, the phenolic profile of calafate, characterized by its high levels of anthocyanins and hydroxycinnamic acids, explains the significant phenolic contribution observed in F3, in line with previous studies that report its high phenolic content [1].

Finally, although F1 and F2 contain the same proportion of calafate and pseudocereals, F1 had a higher total phenolic content, which could be attributed to its higher proportion of oats (120 g versus 60 g in F2). Although oats are not as rich in polyphenols as other functional ingredients, they have been reported to contain various phenolic compounds, including avenanthramides, a group of hydroxycinnamic acid amide conjugates unique to oats, which possess potent antioxidant and anti-inflammatory properties. Additionally, oats contain phenolic acids (such as ferulic, caffeic, p-coumaric, and sinapic), flavonoids, and benzoic acids, which may complement TPC [39]. Overall, the results show that the significant differences in TPC between the formulations are not random but directly reflect the proportions of key ingredients, such as calafate and pseudocereals, thereby confirming the relevance of formulation design in enhancing the functional character of cereal bars.

### 3.3. Determination of Antioxidant Capacity

The antioxidant capacity of the different formulations, evaluated using the DPPH assay, is presented in Table 3. The results showed significant differences between samples (*p* < 0.05), following a trend consistent with the total phenolic content values. The observed variations indicate that the proportion of bioactive ingredients directly influences the antioxidant potential of the cereal bars.

Formulation F3 showed the highest antioxidant capacity (2181.68 ± 59.01 µM TE/100 g), significantly higher than the other formulations (*p* < 0.05). This result confirms the strong association between the presence of phenolic compounds and antioxidant activity. The difference observed can be attributed to the higher proportion of calafate (48 g) compared to F1 and F2 (24 g each, Table 1), as this Patagonian fruit has been widely recognised for its high concentration of anthocyanins and phenolic acids, compounds with high reducing power and free radical scavenging capacity [34].

The strong positive correlation between total phenolic content and antioxidant capacity observed in all formulations (F3 > F1 > F2 > F4) confirms that phenolic compounds are the main contributors to the antioxidant activity of these cereal bars. This relationship is well established in scientific literature, where anthocyanins and other flavonoids have been shown to have a high capacity to scavenge free radicals through multiple mechanisms, including hydrogen atom transfer (HAT) and single electron transfer (SET). The DPPH assay used in this study primarily measures the SET mechanism, whereby phenolic compounds donate electrons to stabilize the DPPH radical. However, it is important to recognize that in vitro antioxidant capacity, while useful for comparative and quality control purposes, does not necessarily predict in vivo bioactivity due to factors such as bioaccessibility, bioavailability, metabolism, and tissue distribution. Anthocyanins, despite their potent in vitro activity, tend to have low bioavailability extensive metabolism, and a relatively short plasma half-life. Nevertheless, both the original compounds and their metabolites have demonstrated biological effects in cellular and animal models, suggesting that even low systemic concentrations may confer health benefits [8,40].

Formulation F4 had the lowest antioxidant capacity (1703.12 ± 32.78 µM TE/100 g), which is consistent with its composition, based exclusively on oats (240 g) and without the inclusion of calafate or pseudocereals, key ingredients in the provision of bioactive compounds. These results are consistent with studies showing a dose–response effect in bars enriched with berries, in which doubling the amount of fruit generates significant increases in antioxidant capacity [38]. Likewise, the positive correlation between total phenolic content and antioxidant capacity, widely documented in functional foods [41], is clearly observed in all formulations.

Finally, it should be noted that calafate berry extracts not only have high antioxidant capacity, but they have also been shown to decrease the production of reactive oxygen species (ROS) in endothelial cell models and protect low-density lipoproteins (LDL) from oxidation, a key process in the progression of atherosclerosis [1,34]. This bioactive effect enhances the relevance of calafate as a functional ingredient of interest in food formulations

### 3.4. Theoretical Nutritional Composition of Cereal Bars

The nutritional components listed in Table 4 are theoretical estimates derived from standard food composition tables [20,21] and declared formulation amounts, rather than from analytical determinations. Accordingly, the data should be interpreted as formulation-level estimates useful for comparing treatments, not as laboratory-measured label values. Theoretical calculations assume additive contributions from individual ingredients and do not consider potential nutrient losses during processing, such as thermal degradation or leaching. They also do not consider matrix interactions that may affect nutrient bioavailability, such as Maillard reactions or the formation of protein-polyphenol complexes, or changes in nutrient extraction capacity resulting from structural modifications during baking. Therefore, the actual nutrient content and bioavailability in final products may differ from the calculated values. Direct analytical validation using standardized methods is required for commercial labeling and confirmation of the nutritional claims presented in this document. However, theoretical calculations provide relevant comparative information for formulation optimization and allow for a preliminary assessment of the nutritional potential of different formulation strategies. Future research should include comprehensive analytical determinations of macronutrients and micronutrients, as well as bioavailability studies, to validate these theoretical estimates and evaluate the impact of processing conditions on nutritional quality.

Table 4 provides the estimated nutritional composition of the different cereal bar formulations, including energy content and macro- and micronutrient intake, expressed both per 100 g of product and per individual serving, together with their respective percentage of the recommended daily value.

#### 3.4.1. Macronutrients

The highest carbohydrate content is found in F4 (14.3 g), which, as with saturated fats, is due to the higher amount of oats, quinoa, and amaranth, followed by F3 (12.03 g), F1 (11.83 g), and F2 (10.5 g) (Table 4). In terms of fiber content, we can see that the highest amount is found in F4 (1.33 g), followed by F3 (0.97 g), F1 (0.88 g), and F2 (0.6 g). The soluble and insoluble dietary fiber in these pseudocereals is classified as a pectic polysaccharide. The second-most prominent dietary fiber is called xyloglucans, present in whole amaranth and quinoa grains. The evaluation of fiber content is important because high consumption of whole grains or cereal fiber may be associated with a lower risk of cardiovascular disease and lower LDL (low-density lipoprotein) and total cholesterol levels [42].

Amaranth, quinoa, and oats have a high fiber content [43], but this value may not have been achieved because the quinoa and amaranth grains used were popcorn types, which reduce their fiber and protein content. Additionally, considering the reduction in nutrient density associated with the expansion process, it is reasonable to suggest that the use of whole or minimally processed pseudocereals could appreciably enhance the fiber and protein content in future formulations. Recent studies indicate that whole and even germinated grains retain higher proportions of insoluble fiber, proteins, and associated micronutrients, suggesting a potential nutritional improvement if expanded grains were replaced with their whole-grain counterparts [44]. Exploring this reformulation strategy in future work may contribute to optimizing the nutritional and functional profile of the bars. For protein (Table 4), the highest contribution was from F4 (1.92 g), followed by F3 (1.1 g), F1 (1.07 g), and F2 (0.63 g), matching the result obtained for fiber from the grains used. Although the results for macronutrients are not outstanding, the bars are designed as a snack or light meal, which should be complemented with other foods during the day. Therefore, consuming the bar would undoubtedly provide energy and healthy fats that are important for daily functioning.

#### 3.4.2. Micronutrients

Regarding micronutrient content, the results showed that F4 provided the highest calcium content (6.77 mg), followed by F3 (4.41 mg), F1 (4.24 mg), and F2 (2.89 mg). For magnesium, the same rankings were maintained: F4 (20.05 mg), F3 (11.71 mg), F1 (11.6 mg), and F2 (7.32 mg). The ingredients that provide the most calcium and magnesium are cereals such as oats [45] and pseudocereals such as quinoa and amaranth [4], which are found in these proportions in the formulations. Phosphorus and zinc follow the same order in quantities according to the formulas.

For Vitamin C, we can find variations due to the incorporation of Calafate in greater quantities in F3 (2.2 mg), F1 and F2 (1.38 mg), and F4 (0.56 mg). The decrease in Vitamin C is proportional to the decrease or absence of Calafate. B vitamins (B1, B2, B3, B5, and B6) follow the same order as calcium, magnesium, phosphorus, and zinc due to the amount of pseudocereals (quinoa and amaranth) used in their formulations, as mentioned by D’ amaro et al. [43] It is also important to consider that the afore mentioned pseudocereals not only provide micronutrients and energy but are also useful as adjuvants in the prevention and treatment of patients with diabetes mellitus [46], lowering total cholesterol, LDL cholesterol, and triglycerides in people with diabetes and a Body Mass Index (BMI) > 25 [47].

However, health is not only determined by the type of food, but also by the geographical area where it is grown, which influences its nutritional content and bioactive components [9]. which is also related to food sovereignty, where the consumption of native foods such as quinoa, amaranth, and calafate (included in the cereal bar formulations presented) is promoted, thus contributing to reducing food insecurity [48].

In addition, in recent years, there has been growing interest in the consumption of cereal bars as a healthy alternative to traditional snacks such as biscuits, crisps, or products with a high refined sugar content. Several studies report a trend towards replacing these products with fruit, dairy products, and bars formulated with natural and functional ingredients, with the aim of improving diet quality and promoting health benefits [15]. This approach is consistent with the principles that guided the formulation development evaluated in this study.

### 3.5. Sensory Evaluation of Cereal Bars

The sensory evaluation of the four cereal bar formulations was carried out by a panel of evaluators using hedonic, and structured scales to determine the acceptability of the product, and its sensory attributes.

#### 3.5.1. Acceptance, Preference, and Purchase Intention

Consumer-oriented sensory evaluation (acceptability, preference, and purchase intention) is key in food development because it allows for estimating product acceptance and commercial performance [49]. Table 5 summarizes the results for acceptability, acceptability index, preference, and purchase intention obtained from the sensory evaluation. Formulation F2 obtained the highest overall acceptance score (7.06 ± 1.73), significantly higher than F1 (6.06 ± 1.24) and slightly higher than F3 and F4, between which no statistical differences were observed. This result was also reflected in the acceptability index, where F2 reached 78.41%, the highest value among all formulations. F3 and F4 showed intermediate values (72.70% and 73.01%, respectively), while F1 had the lowest index (67.30%), indicating lower hedonic acceptance by the panel. These acceptability values are encouraging from a commercial perspective, as they exceed the minimum threshold of 70% reported in the literature as an indicator of market viability for new food products. Ref. [50] established that products with acceptability indices above 70% have a high probability of commercial success. In this context, F2, F3, and F4 demonstrated favorable commercial potential, while F1 was slightly below this threshold.

In terms of preference, F2 stood out again (7.57 ± 1.38), significantly outperforming F1 (6.60 ± 1.06), and with a non-significant advantage over F3 (7.06 ± 1.26) and F4 (6.80 ± 1.64). The superiority of F2 in acceptance and preference aligns with its intermediate hardness (Table 2), which likely delivers a balanced soft–chewy/crunchy bite. The replacement of part of the oats with expanded pseudocereals probably reduced bulk density and altered the matrix microstructure, yielding a lighter texture and a more appealing flavor profile. By contrast, F1, with the lowest hardness (Table 2), may have been softer and less crisp for the consumer, resulting in a less distinctive sensory profile. Thus, preference appears to peak at moderate hardness, consistent with the instrumental data in Table 2.

However, purchase intention did not show statistically significant differences between the formulations (*p* > 0.05), with values between 3.57 ± 1.01 (F1) and 4.03 ± 1.15 (F2). This indicates that, although F2 obtained the highest sensory rating, there was a favorable predisposition among consumers towards purchasing the product in general, suggesting commercial potential for both formulations. Purchase intention is a crucial predictor of consumer behavior in real market conditions. The values recorded in this study exceed the midpoint of the scale (3.0), which shows a positive disposition towards purchase, in accordance with what was reported by [51] for structured scales of purchase intention in food products. Although no statistically significant differences were identified in this parameter, the numerical trend favors F2, reinforcing its position as the formulation with the greatest commercial potential.

The absence of significant differences in purchase intention between formulations indicates that additional factors, beyond intrinsic sensory characteristics, may influence the purchase decision. These factors include price, nutritional information on labelling, health claims, and marketing strategies. Research by Cardello et al. [52] has shown that purchase intention for functional products such as cereal bars is strongly influenced by perceived health benefits and consumer familiarity with the ingredients used.

The incorporation of functional ingredients such as quinoa, amaranth, and calafate in the formulations studied represents a competitive advantage in today’s market, where there is a growing demand for natural, nutritious foods with recognizable ingredients. In this regard, after considering the competitive potential provided by functional ingredients, it is important to note that the sensory evaluation performed corresponds to a specific moment in the product’s life cycle. The stability of organoleptic characteristics during storage is a critical aspect that should be evaluated in future studies, particularly considering the presence of components susceptible to change, such as unsaturated fatty acids in sunflower oil and bioactive compounds in fruits [53], emphasized the importance of conducting periodic sensory evaluations during the shelf life of medium-moisture products such as cereal bars to ensure that consumer acceptability is maintained throughout the marketing period.

#### 3.5.2. Sensory Attributes

The results of the sensory attribute analysis, presented in Figure 2, showed significant differences between the formulations evaluated. The radar chart indicates that formulation F4 obtained the highest scores for appearance and flavor, establishing itself as the option with the most favorable sensory profile in these aspects. The addition of 240 g of oats in F4 probably contributed to a more robust and textured appearance, which was perceived as attractive by the evaluators. In contrast, F2 scored higher in taste and aroma than F1 and F3, which can be attributed to the higher proportion of pseudocereals (amaranth and quinoa) present in this formulation, which contribute distinctive organoleptic characteristics. On the other hand, F3 received the lowest scores for texture and aroma, which could justify its lower overall acceptance (Table 5), despite its more outstanding nutritional and functional profile.

The inclusion of calafate increased the phenolic content and antioxidant capacity, but it could also introduce organoleptic challenges, especially in formulations such as F3, which had a higher concentration of bioactive compounds. The darker color and astringency associated with anthocyanin-rich ingredients probably influenced the perception of flavor and texture. On the other hand, the better performance of F2 is attributed to a more balanced matrix between oats and pseudocereals, with a moderate calafate content, which allowed for a more favorable sensory compromise, although determining optimal levels requires additional optimization studies.

Color showed uniform scores among the formulations, with F1 and F2 obtaining slightly higher values. This parameter is mainly determined by the Maillard reaction during thermal processing and by the natural pigments of the ingredients, particularly the anthocyanins present in calafate. The lower amount of oats in F2 (60 g) allowed for greater expression of the color provided by the calafate and pseudocereals.

The texture and aroma presented homogeneous values among all formulations, with scores between 3.54 and 3.89, indicating moderate acceptance of these attributes by the sensory panel. Texture is a critical parameter in cereal bars, as it requires a balance between firmness and chewiness. The textural characteristics observed have important implications for consumer acceptance and product quality. While firmer textures (higher maximum force) may indicate better structural integrity and stability during storage, high hardness products may negatively impact consumer acceptance due to difficulty in chewing. The instrumental evaluation of texture using uniaxial compression (Table 2) revealed a significant correlation with the sensory evaluation results. Formulation F2, which had an intermediate hardness (close to 12 N), obtained the highest sensory scores (Figure 2), while F4, which had the highest value (27.63 N) in uniaxial compression, had the lowest score in the texture parameter (Figure 2). This pattern suggests that consumer taste peaks with an intermediate texture (neither too soft nor too hard), a behavior commonly observed in cereal/nutritional bars, where excessive hardening reduces acceptability [30].

About aroma, no significant differences were identified between formulations F1, F2, and F4, whose scores remained within a moderate range (Figure 2). In contrast, F3 had the lowest score in this aspect, which could be attributed to the higher concentration of calafate. This concentration may have caused an imbalance in the product’s aroma profile, as the volatile compounds characteristic of this Patagonian forest fruit competed with the aromas of other key ingredients such as honey, toasted pseudocereals, and cranberries.

Overall, the results suggest that F2 provides the most appropriate balance between technological functionality and sensory acceptance among the formulations analyzed. Although formulations enriched with higher amounts of calafate, such as F3, have biofunctional advantages, possible compromises in hedonic perception must be considered when developing products intended for mass consumption.

Several limitations of the present sensory study should be recognized and addressed in future research. First, the panel comprised untrained consumers (n = 35), which is sufficient for affective testing and commercial viability assessment but restricts the depth of sensory characterization. Employing trained descriptive sensory panels and methodologies such as quantitative descriptive analysis (QDA) would yield more detailed and objective profiles of sensory attributes, enabling precise identification of specific flavor notes, texture characteristics, and aromatic compounds that influence consumer preferences. Second, the study assessed the fresh bars immediately after production without evaluating changes during storage, which could provide valuable insights into dynamic sensory perception. Incorporating these methodological enhancements, including sensory testing over the product’s shelf life, would offer a more comprehensive understanding of product performance and inform optimization strategies for commercial development.

## 4. Conclusions

The study confirmed that incorporating Andean pseudocereals and calafate berries into cereal bars increases their functional, nutritional, and sensory properties. In terms of texture, using pseudocereals in a complementary proportion to oats produced a softer matrix, while adding calafate berries reinforced structural cohesion without affecting palatability. In terms of bioactivity, formulation F3, with a higher proportion of calafate, showed the highest content of phenolic compounds and antioxidant capacity, highlighting the key functional role of this Patagonian fruit. However, even formulations with lower amounts of calafate showed high antioxidant capacity attributed to the incorporated pseudocereals. In terms of nutritional profile, the bars provide an adequate supply of energy, carbohydrates, and fats, as well as vitamins and minerals such as calcium, magnesium, iron, zinc, B vitamins, and vitamin C. However, the protein and fiber content were not significant, possibly due to the use of quinoa and amaranth in pop format instead of whole grains, an aspect that could be optimized in future formulations. By integrating results on nutritional composition, phenolic content, antioxidant capacity, texture, and sensory acceptance, formulation F2 emerges as the optimal formulation, offering the best balance between functional attributes and organoleptic characteristics. In addition, formulation F2 was the most accepted, with 78.41% of the panel’s preference. Overall, the results indicate that these cereal bars are a suitable alternative as a mid-morning or mid-afternoon snack, combining bioactive compounds relevant to the prevention of chronic non-communicable diseases, good sensory acceptance, and the valorization of indigenous ingredients, thus contributing to the strengthening of food sovereignty.

Although the results are promising, this study has some limitations. The bars were evaluated only under laboratory conditions and with a small sensory panel. Their shelf life and effects after in vitro digestion were not determined. In addition, the use of pseudocereals in “pop” format limited the potential increase in protein and fiber that could be achieved with whole grains or flours. Therefore, it is recommended that future studies optimize the type and proportion of pseudocereals and calafate, evaluate the stability of bioactive compounds and sensory attributes during storage, and analyze the bioaccessibility and metabolic impact of these bars in larger consumer groups.

## Figures and Tables

**Figure 1 foods-14-04127-f001:**
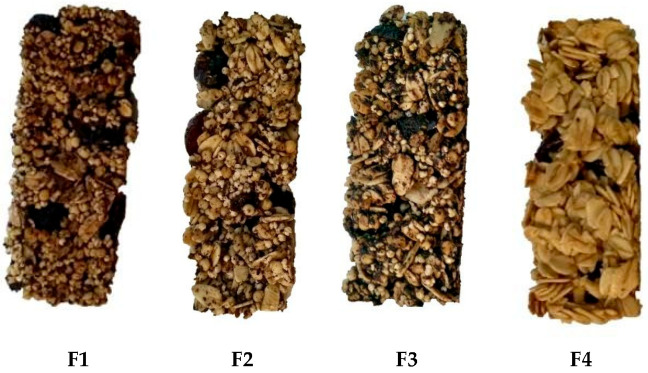
Images of the different formulations of cereal bar: F1—Baseline cereal bar with balanced ingredients; F2—Pseudocereal-enriched bar; F3—High-calafate bar; F4—Oats-only control.

**Figure 2 foods-14-04127-f002:**
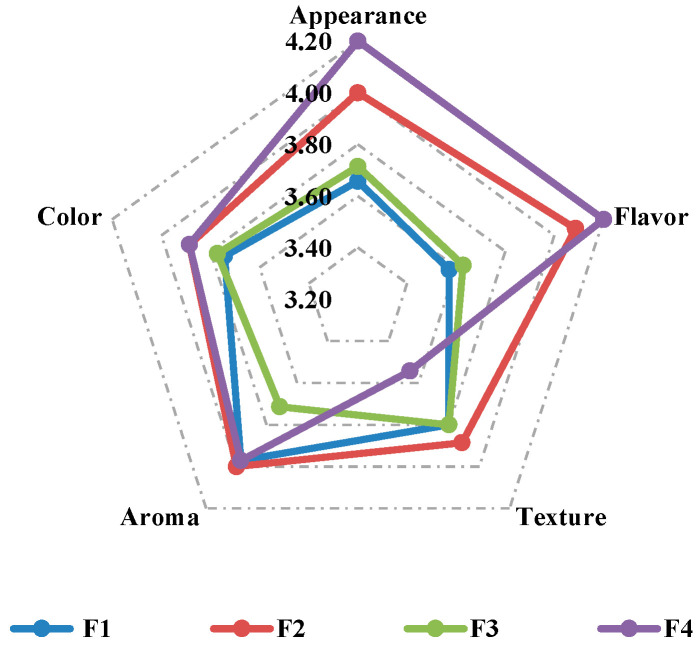
Sensory attributes evaluated for the different formulations of formulated cereal bars. Each line corresponds to a specific formulation, and the scores reflect the mean values assigned by the sensory panel using a 5-point hedonic scale (1 = dislike extremely, 5 = like extremely).

**Table 1 foods-14-04127-t001:** Raw material and proportion used in the different formulations of cereal bars.

Ingredients (g)	Formulations			
	F1	F2	F3	F4
Oats	120	60	120	240
Cranberries	80	80	80	80
Amaranth popcorn	14	21	14	
Quinoa popcorn	14	21	14	
Honey	170	170	170	170
Sunflower oil	10	10	10	10
Calafate	24	24	48	

The table presents the quantity (g) of each ingredient used in formulations F1, F2, F3, and the control (F4). Variations among formulations were designed to evaluate the influence of different ingredients, such as calafate and pseudocereal popcorn, on the nutritional and functional properties of the bars.

**Table 2 foods-14-04127-t002:** Peak compression force (hardness, N) for the texture evaluation of formulated cereal bars.

Formulations	Maximum		
	Force (N)		
F1	7.31	±	1.24 ^a^
F2	11.96	±	2.04 ^b^
F3	20.27	±	2.33 ^c^
F4	27.63	±	0.18 ^d^

Hardness was defined as the peak force reached during a 10 mm compression. Data are presented as mean ± standard deviation (*n* = 10). Different letters in columns (a, b, c, and d) indicate statistically significant differences. Values differ significantly at *p* < 0.05.

**Table 3 foods-14-04127-t003:** Total phenol content and antioxidant capacity in different formulations of cereal bars.

Formulations	Total Phenolic Content			Antioxidant Capacity		
	(mg GAE/100 g Sample)			(µM TE/100 g Sample)		
F1	264.67	±	9.76 ^a^	1943.19	±	32.46 ^b^
F2	231.18	±	10.63 ^b^	1987.37	±	63.04 ^b^
F3	403.49	±	15.84 ^c^	2181.68	±	59.01 ^c^
F4	141.34	±	9.73 ^d^	1703.12	±	32.78 ^a^

Data are presented as mean ± standard deviation (n = 3). Different letters in columns (a, b, c, and d) indicate statistically significant differences. Values differ significantly at *p* < 0.05.

**Table 4 foods-14-04127-t004:** Nutritional components of different formulations of cereal bars.

Compounds and Percentage	Formulations											
	F1			F2			F3			F4		
	Serving Size	100 g	Serving Size (%)	Serving Size	100 g	Serving Size (%)	Serving Size	100 g	Serving Size (%)	Serving Size	100 g	Serving Size (%)
Calories (Kcal)	55.68	278.4	2.80	46.8	234.1	2.30	56.12	280	2.81	72.94	364.7	3.65
% Daily Value *												
Total Fat (g)	0.91	4.60	0.41	0.75	3.80	0.34	0.92	4.60	0.41	1.23	6.20	0.55
Saturated Fat (g)	0.31	1.50	0.14	0.36	1.80	0.16	0.33	1.60	0.15	0.18	0.90	0.08
Monounsaturated Fat (g)	0.22	1.10	0.10	0.17	0.80	0.08	0.22	1.10	0.09	0.33	1.70	0.15
Polyunsaturated Fat (g)	0.46	2.30	0.21	0.40	2.0	0.18	0.46	2.30	0.21	0.58	2.90	0.26
Sodium (mg)	0.69	3.50	NA	0.72	3.60	NA	0.71	3.60	NA	0.61	3.10	NA
Total Carbohydrate (g)	11.83	59.20	2.36	10.5	52.5	2.10	12.03	60.2	2.40	14.3	71.5	2.86
Dietary Fiber (g)	0.88	4.40	0.08	0.60	3.0	0.06	0.97	4.90	0.09	1.33	6.60	0.13
Total Sugars (g)	6.79	34.0	1.36	6.78	33.9	1.36	6.87	34.4	1.37	6.73	33.7	1.35
Protein (g)	1.07	5.30	0.21	0.63	3.10	0.12	1.10	5.50	0.22	1.92	9.60	0.38
% RDI **												
Calcium (mg)	4.24	21.20	0.50	2.89	14.5	0.40	4.41	22.0	0.60	6.77	33.8	0.80
Magnesium (mg)	11.60	58.0	3.70	7.32	36.6	2.30	11.71	58.60	3.70	20.05	100.2	6.40
Phosphorus (mg)	32.96	164.8	4.10	19.92	99.6	2.50	33.18	165.9	4.10	58.83	294.2	7.40
Potassium (mg)	12.19	60.90	0.40	13.89	69.5	0.50	13.43	67.20	0.40	7.53	37.7	0.30
Zinc (mg)	0.40	2.0	3.30	0.30	1.50	2.50	0.40	2.0	3.30	0.46	2.30	3.80
Vitamin C (mg)	1.38	6.90	0.20	1.38	6.90	0.20	2.20	11.0	1.80	0.56	2.80	0.13
Thiamine B1 (mg)	0.04	0.20	3.25	0.02	0.10	1.82	0.04	0.20	3.25	0.07	0.40	6.10
Riboflavin B2 (mg)	0.01	0.10	0.83	0.01	0.10	0.83	0.01	0.10	0.83	0.02	0.10	1.60
Niacin B3 (mg)	0.08	0.40	0.50	0.05	0.30	0.40	0.08	0.40	0.50	0.12	0.60	0.80
Pantothenic acid B5 (mg)	0.10	0.50	2.0	0.07	0.30	1.40	0.10	0.50	0.20	0.17	0.80	3.30
Pyridoxine B6 (mg)	0.02	0.10	1.20	0.01	0.10	1.10	0.02	0.10	1.20	0.02	0.10	1.20
Folate B9 (µg)	3.82	19.10	1.0	2.51	12.60	0.60	3.82	19.10	1.0	6.42	32.10	1.60

Values are theoretical estimates calculated from standard composition tables (INTA [20]; USDA data [21]). No laboratory assays were performed for nutrient quantification. * The % Daily Value (DV) tells you how much a nutrient in a serving of food contributes to a daily diet. 2000 calories a day is used for general nutrition advice; ** Vitamin and mineral contributions were expressed as a percentage of the Recommended Daily Intake (RDI), reflecting the daily nutrient requirements; NA: Not Available.

**Table 5 foods-14-04127-t005:** Sensory evaluation of cereal bars formulations based on acceptance, acceptability index, preference, and intention of purchase.

Formulations	Acceptance			Acceptability Index (%)	Preference			Intention to Purchase		
F1	6.06	±	1.24 ^a^	67.30 ^a^	6.60	±	1.06 ^a^	3.57	±	1.01 ^a^
F2	7.06	±	1.73 ^b^	78.41 ^b^	7.57	±	1.38 ^b^	4.03	±	1.15 ^a^
F3	6.54	±	1.77 ^ab^	72.70 ^c^	7.06	±	1.26 ^ab^	3.83	±	1.15 ^a^
F4	6.57	±	1.84 ^ab^	73.01 ^c^	6.80	±	1.64 ^a^	3.63	±	1.31 ^a^

Acceptance was assessed using a 9-point FACT scale (1 = I would only eat this if I had to, 9 = I would eat this whenever I had the chance). Preference was assessed using a 9-point hedonic scale (1 = dislike extremely, 9 = like extremely). Purchase intention was measured using a 5-point structured scale (1 = definitely would not buy, 5 = definitely would buy). Different letters in columns (a, b, and c) indicate statistically significant differences. Values differ significantly at *p* < 0.05.

## Data Availability

The original contributions presented in the study are included in the article; further inquiries can be directed to the corresponding author.

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
