# Peer review of "Development of Cereal Bars Enriched with Andean Grains and Patagonian Calafate (Berberis microphylla): Nutritional Composition, Phenolic Content, Antioxidant, Textural, and Sensory Evaluation"

_foods, 2025, doi:10.3390/foods14234127_

Round 1

Reviewer 1 Report

Comments and Suggestions for Authors

This article evaluated nutritional composition, antioxidant, texture, and sensory parameters on different types of cereal bars. The methods are straightforward, and my concerns are as below.

  1. What were used as the control and blank in the TPC analysis? How did you ensure that the Folin–Ciocalteu-based TPC results specifically reflect phenolic content, rather than other reducing components that may interfere with the reagent?
  2. What was used as the blank in the DPPH assay, and what was the concentration range of the standard curve?
  3. The font size of footnotes in all tables should be decreased.
  4. Why did sample F1, which showed significantly higher TPC, exhibit similar antioxidant activity compared to F2?
  5. Why were the serving sizes different among these samples?
  6. What are the differences in nutrient composition across these samples?

Author Response

Ref.: Manuscript ID: Foods- 3992770

Responses to Reviewers' comments

Summary

Thank you very much for handling our manuscript entitled “Development of cereal bars enriched with Andean grains and Patagonian calafate (Berberis microphylla): nutritional compo-sition, phenolics, antioxidant, textural, and sensory evaluation” (Manuscript ID: Foods-3992770). We greatly appreciate the opportunity that you have given us to review further this manuscript.

Thank you very much to the reviewers for the constructive comments. The authors have done their best to attend to all the comments, either by rewriting or amending the manuscript or justifying the action. Based on the valuable remarks given to us, the manuscript was redrafted in several sections. We trust the manuscript is more fluent and understandable now.

We are pleased that the referees found our study interesting, and we appreciate their comments, which have helped us to improve our manuscript.

Please find below our reply to these comments. All changes in the revised manuscript were highlighted by using red text. We hope that you will find our manuscript worthy of publication in Foods.

Reviewer #1:

This article evaluated nutritional composition, antioxidant, texture, and sensory parameters on different types of cereal bars. The methods are straightforward, and my concerns are as below.

Comment 1: What were used as the control and blank in the TPC analysis? How did you ensure that the Folin–Ciocalteu-based TPC results specifically reflect phenolic content, rather than other reducing components that may interfere with the reagent?

Response 1: We appreciate your comment and the opportunity to clarify this point.

In our research, TPC analysis was performed using a microplate reader, so we distinguish between the terms “blank” and “control” as follows:

Assay blank: In our protocol, we only used one blank, which corresponds to the mixture of reagents without a sample, where the sample volume (5 µL) was replaced by the extraction solvent (ethanol/water 50:50, v/v). That is, each blank well contained: extraction solvent + Folin–Ciocalteu reagent + Naâ‚‚CO₃ solution + distilled water, in the same volumes and incubation conditions as the samples. The blank was used to adjust the absorbance zero in the microplate reader and correct any signal due exclusively to the reagents and solvent.

Control in the context of the analysis: In our protocol, we did not use an additional control sample (e.g., a specific formulation or a sample without phenolic extract) measured in parallel as a “control.” What we used was a reference from the assay, where the standard used was gallic acid, to prepare the calibration curve. All absorbance readings of the samples corrected for the blank were interpolated on this curve and expressed as mg of gallic acid equivalents (GAE) per 100 g of sample.

Regarding the specificity of the Folin–Ciocalteu reagent, we are very clear that this method evaluates the total reducing capacity and may respond to other reducing compounds besides phenolic compounds, for example, reducing sugars or ascorbic acid. To reduce this interference and ensure that the signal mainly reflects the total phenolic content, we considered the following aspects:

  • A 50:50 (v/v) ethanol/water mixture was used, which is widely used to extract phenolic compounds from cereal and fruit matrices.
  • The extraction conditions were moderate (room temperature, controlled times), seeking to preserve the extracted phenolic compounds.
  • In the matrix studied (cereal bars), the main contributors with reducing potential are calafate (rich in anthocyanins and other phenolic compounds) and pseudocereals, whose bioactive fractions have been mainly described as phenolic.
  • We found a consistent link between TPC values and antioxidant capacity measured by DPPH in all samples, which supports that polyphenols are mainly responsible for the Folin–Ciocalteu reagent’s response.

Comment 2: What was used as the blank in the DPPH assay, and what was the concentration range of the standard curve?

Response 2: In our DPPH assay measurement, we did not use an independent blank (i.e., a well without the DPPH free radical). Since the standard curve was constructed based on the percentage of inhibition relative to the DPPH control, then a control was used.

This control corresponded to the DPPH solution in ethanol (solvent) without a sample or antioxidant standard, only with the working solvent. The absorbance of this control was used as a reference value in the following equation:

Where Abs control is the absorbance of the control (mixture DPPH + ethanol) and Abs sample is the absorbance of the DPPH + sample mixture (or Trolox standard). In this way, any contribution from the solvent and the DPPH reagent is included in the Abs control and canceled out in the calculation of the percentage inhibition, so no additional blank was necessary.

As for the standard curve, Trolox solutions in ethanol were prepared in the range of 0 to 50 μM. For each concentration, the percentage of inhibition was calculated with respect to the same DPPH control, and these values were used to construct the calibration curve, from which the antioxidant capacity of the samples was expressed as μM equivalents of Trolox per 100 g of sample.

Comment 3: The font size of footnotes in all tables should be decreased.

Response 3: We appreciate your comment. In the revised version of the manuscript, the font size of the footnotes in all tables has been reduced, as suggested.

Comment 4: Why did sample F1, which showed significantly higher TPC, exhibit similar antioxidant activity compared to F2?

Response 4: We appreciate your observation. In our study, formulation F1 had a higher total phenolic compound (TPC) content than F2, while both showed statistically similar antioxidant capacity as measured by DPPH, as they are in the same homogeneous group (see Table 3). This apparent discrepancy can be explained by several factors:

1) Although the difference in TPC between F1 and F2 is statistically significant, it is relatively moderate and falls within a concentration range in which the DPPH assay tends to reach a near-plateau response. Under these conditions, small increases in TPC do not necessarily translate into proportional increases in DPPH radical inhibition.

2) The Folin–Ciocalteu method quantifies the overall reducing capacity of the sample and responds to a broad spectrum of phenolic compounds as well as other non-phenolic reducing compounds. In contrast, the DPPH assay is more sensitive to those compounds capable of donating hydrogen or electrons particularly efficiently. Therefore, F1 could contain a greater amount of phenols that react with the Folin–Ciocalteu reagent but contribute less to DPPH radical scavenging compared to the phenolic profile of sample F2.

3) The differences in the proportion of oats, expanded pseudocereals, and calafate between F1 and F2 modify the structure of the food matrix.

These modifications can influence the release, solubility, and accessibility of compounds in the extraction solvent , thereby affecting their interaction with the DPPH radical, and resulting in similar antioxidant capacity values despite differences in TPC.

However, when all formulations (F1–F4) are evaluated together, a consistent trend between TPC and antioxidant capacity is maintained, as formulation F3 (with the highest level of calafate) has the highest values for both determinations, while F4 (without calafate or pseudocereals) has the lowest values.

Comment 5: Why were the serving sizes different among these samples?

Response 5: Thank you for your comment. As indicated in Line 137 of the Materials and Methods section, each formulation was distributed evenly in silicone molds previously greased with sunflower oil, measuring 11 × 4 × 2 cm, so that all bars were the same size and weight.

In Figure 1, although they appear to be different sizes, this is due to the photograph, which may give the impression of differences in the size of the bars due to the angle of the shot; however, they were all made with the same volume of dough and baked in the same standardized molds.

Comment 6: What are the differences in nutrient composition across these samples?

Response 6:  Dear reviewer, we accept your recommendation in sections 3.4.1 and 3.4.2. The differences in nutritional composition between the formulations are presented implicitly, as the narrative focuses on highlighting those nutrients whose contribution is relevant and, at the same time, differentiates between the different formulations. These variations respond directly to the specific proportions of oats, pseudocereals (quinoa and amaranth), and calafate incorporated into each sample, which explains the differences observed in both macronutrients (mainly carbohydrates, fiber, and proteins) and micronutrients (minerals and vitamins). Thus, the discussion emphasizes how the formulation design determines the nutritional profile of each bar, allowing for a clear quantitative comparison between samples.

Reviewer 2 Report

Comments and Suggestions for Authors

The manuscript titled “Development of cereal bars enriched with Andean grains and Patagonian calafate (Berberis microphylla): nutritional composition, phenolics, antioxidant, textural, and sensory evaluation”, deals with an interesting topic, highlighting health benefits of pseudocereals and calafate when they are included in nutritional bars.

Authors discussed the preparation of out-based nutrition bars and the influence of addition of amaranthus, quinoa and calafate powder on its structure, texture, antioxidant capacity and nutritional properties. The topic of the work should be of interest to the readership of the foods journal. However, the work must be improved by the inclusion of additional information in some sections. Errors exist in the manuscript, which are also required to be corrected, for example, in the title of the manuscript, the scientific name of Patagonian calafate is written as “Berberis Microphylla”, “M” should be written in lowercase letter “m”. Scientific names should be written in italics.

Abstract

Authors include formulations as abbreviations “F1, F2, F3 and F4” in the abstract section, making difficult to follow. For a better understanding, I suggest changing these abbreviations to the main characteristics of each formulation. “The incorporation of calafate enhances the phenolic and antioxidant profile in F3, while an intermediate hardness linked to the greater use of expanded pseudocereals”.

Introduction

Although introduction includes the key points of the manuscript, it is hard to see the innovation in this work, except for calafate berries, the raw materials used to produce the bars are widely available.

Materials and methods

Please explain how you select these four formulations.

The description of the experiments is not precise, for instance, in the description of texture analysis authors do not specify the geometry used for this analysis.

I suggest that proximal analysis must be done to compare with theoretical data.

Results and discussion

I found the discussion of the manuscript most descriptive with expected results. However, it is necessary to include a discussion based on scientific evidence including more references.

Lines 294-296. It is expected that the total phenol compounds and the antioxidant capacity will increase as a function of berries’ concentration in nutritional bars. Please explain the innovation of adding blue and calatate berries.

The content of proteins and fiber were higher in F4, therefore, how the inclusion of other pseudocereals is justified.

Conclusion

In the conclusion section, major improvements should be made, including the optimal formulation.

Overall, I believe the manuscript needs some improvements before publication in “Foods”.

Author Response

Ref.: Manuscript ID: Foods- 3992770

Responses to Reviewers' comments

Summary

Thank you very much for handling our manuscript entitled “Development of cereal bars enriched with Andean grains and Patagonian calafate (Berberis microphylla): nutritional compo-sition, phenolics, antioxidant, textural, and sensory evaluation” (Manuscript ID: Foods-3992770). We greatly appreciate the opportunity that you have given us to review further this manuscript.

Thank you very much to the reviewers for the constructive comments. The authors have done their best to attend to all the comments, either by rewriting or amending the manuscript or justifying the action. Based on the valuable remarks given to us, the manuscript was redrafted in several sections. We trust the manuscript is more fluent and understandable now.

We are pleased that the referees found our study interesting, and we appreciate their comments, which have helped us to improve our manuscript.

Please find below our reply to these comments. All changes in the revised manuscript were highlighted by using red text. We hope that you will find our manuscript worthy of publication in Foods.

Reviewer #2:

The manuscript titled “Development of cereal bars enriched with Andean grains and Patagonian calafate (Berberis microphylla): nutritional composition, phenolics, antioxidant, textural, and sensory evaluation”, deals with an interesting topic, highlighting health benefits of pseudocereals and calafate when they are included in nutritional bars.

Authors discussed the preparation of out-based nutrition bars and the influence of addition of amaranthus, quinoa and calafate powder on its structure, texture, antioxidant capacity and nutritional properties. The topic of the work should be of interest to the readership of the foods journal. However, the work must be improved by the inclusion of additional information in some sections.

Comment 1: Errors exist in the manuscript, which are also required to be corrected, for example, in the title of the manuscript, the scientific name of Patagonian calafate is written as “Berberis Microphylla”, “M” should be written in lowercase letter “m”. Scientific names should be written in italics.

Response 1: Thank you for your comment. In the revised version, we have corrected the scientific name to Berberis microphylla (with a lowercase “m” and in italics) in the title and throughout the manuscript, and we have standardized the format of the other scientific names.

Comment 2: Abstract: Authors include formulations as abbreviations “F1, F2, F3 and F4” in the abstract section, making difficult to follow. For a better understanding, I suggest changing these abbreviations to the main characteristics of each formulation. “The incorporation of calafate enhances the phenolic and antioxidant profile in F3, while an intermediate hardness linked to the greater use of expanded pseudocereals”.

Response 2:  We appreciate your suggestion. We agree that the use of abbreviations (F1–F4) in the abstract can make it difficult to read, especially for non-specialist readers. We have therefore revised the abstract and replaced these codes with brief descriptions highlighting the main characteristics of each formulation. Therefore, in the revised version of the abstract, we added a paragraph referring to the four formulations (lines 22–24). In addition, the abbreviations F1–F4 are retained in the body of the manuscript and in the tables, where they are clearly defined and facilitate the concise presentation of the results.

Comment 3:  Introduction: Although the introduction includes the key points of the manuscript, it is hard to see the innovation in this work, except for calafate berries, the raw materials used to produce the bars are widely available.

Response 3: We appreciate your comment and the opportunity to clarify the innovative aspects of our work. We agree that ingredients such as oats, honey, and oil are widely available; however, the novelty of this study is not based solely on the use of calafate, but on the specific combination of ingredients, its territorial approach, and the type of integrated assessment that is carried out.

First, we developed cereal bars formulated exclusively with regionally sourced raw materials, integrating Patagonian calafate with Andean pseudocereals (quinoa and amaranth) and oats, all of which are highly relevant to local biodiversity and agri-food chains in the area. To our knowledge, calafate has not previously been incorporated into cereal bars, and there are few studies on its use in ready-to-eat snacks.

Secondly, systematically compared four formulations with varying proportions of Calafate and expanded pseudocereals. We analyzed how these variations influence nutritional composition, textural parameters, total phenolic compound content, antioxidant capacity, and consumer acceptance.

This integrated evaluation provides new information on the technological feasibility and functional potential of calafate-enriched cereal bars. Therefore, to make this new feature more explicit to readers, we added a paragraph to the Introduction (Lines 102-107).

Comment 4:  Please explain how you select these four formulations.

Response 4: Thanks for your comment.  The four formulations were selected in order to systematically evaluate the effect of calafate and expanded Andean pseudocereals on the technological and functional properties of cereal bars, while also including a more conventional reference product. First, we conducted preliminary tests to determine the appropriate amounts of expanded oats, quinoa, amaranth, and calafate, taking into account the mixture's manageability, the shape of the bars, and their structural stability after baking. Within these ranges, we selected:

F1 as a base cereal bar, with balanced proportions of oats, expanded pseudocereals, and calafate.

F2 is a formulation with lower oat content and a higher proportion of expanded pseudocereals to increase the contribution of Andean grains.

F3 is the formulation with the highest level of calafate and the highest proportion of expanded pseudocereals, with the aim of maximizing the incorporation of regional functional ingredients.

F4, as a control bar made only with oats, without calafate or pseudocereals, represents a more conventional cereal bar.

Thus, the four formulations represent a gradient in the inclusion of calafate and Andean pseudocereals, allowing for analysis of their impact on texture, nutritional composition, phenolic content, antioxidant capacity, and consumer acceptance.

Comment 5: Materials and methods: The description of the experiments is not precise, for instance, in the description of texture analysis authors do not specify the geometry used for this analysis.

Response 5: We appreciate your comment. In the revised version of the manuscript, we have clarified the geometry used in the texture analysis, specifying the portions analyzed. This information was incorporated into the Materials and Methods section.

Comment 6: Materials and methods: I suggest that proximal analysis must be done to compare with theoretical data.

Response 6: We appreciate the reviewer's suggestion and agree that direct proximal analysis would provide valuable nutritional information. However, in the original design of this study, the main focus was on the incorporation of calafate and Andean pseudocereals and their effect on the phenolic compound content, antioxidant capacity, instrumental texture, and sensory acceptance of the cereal bars. In this context, nutritional information was used as a descriptive complement and was estimated from the formulation and official food composition tables, rather than as a main response variable.

Performing complete proximate analyses (moisture, protein, lipids, ash, fiber, and carbohydrates by difference) for the four formulations, with their respective replicates, involves considerable consumption of sample, reagents, and instrumental time that was not contemplated in the experimental plan or in the project resources. Incorporating this battery of analyses now exceeds the scope of the study and is not possible within the deadlines set for manuscript review.

However, we acknowledge this limitation and have made it clearer in the manuscript.

The Materials and Methods section specifies that the reported nutritional composition is based on theoretical values calculated from food composition tables. The Discussion section further notes that these data should be considered approximations and recommends that future research include experimental proximate analysis of the bars to validate and supplement this information.

Comment 7: I found the discussion of the manuscript most descriptive with expected results. However, it is necessary to include a discussion based on scientific evidence including more references.

Response 7:  We appreciate the comment. In the revised version, the Discussion section was strengthened by incorporating more scientific evidence and references, especially in the sections on phenolic compounds and antioxidant capacity, nutritional profile, and sensory results in the context of previous studies.

Comment 8: Lines 294-296. It is expected that the total phenol compounds and the antioxidant capacity will increase as a function of berries’ concentration in nutritional bars. Please explain the innovation of adding blue and calatate berries.

Response 8: Thank you for your important observation. We agree that, from a theoretical point of view, it is expected that the total phenolic compound content and antioxidant capacity will increase as the concentration of berries in a nutritional bar increases. However, the innovation of this work lies not only in demonstrating that “the greater the amount of calafate, the greater the antioxidant capacity,” but also in other important points, such as: a) the fact of introducing calafate into cereal bars, which is a native Patagonian berry that has been little studied and has hardly been used in the manufacture of this type of product. To our knowledge, there are no previous reports of cereal bars enriched with calafate. b) Our results show that calafate, in addition to contributing to territorial identity, can equal or even exceed the phenolic and antioxidant content of conventional berries. In order to make this contribution more explicit, we have added a paragraph to the manuscript (Lines 308-310).

Comment 9: The content of proteins and fiber were higher in F4, therefore, how the inclusion of other pseudocereals is justified.

Response 9: Dear reviewer, the literature indicates that both grains and pseudocereals are important sources of protein and dietary fiber, especially when used in their whole grain form. In this study, we chose to use them in a “popped” (inflated) form due to their organoleptic characteristics and ease of incorporation into the formulations developed. When comparing these formulations with F4 (control bar), which did not contain pseudocereals but did contain a high proportion of rolled oats, a higher protein and fiber content was observed, consistent with the literature and reaffirming the role of oats in improving the nutritional profile of the product. This aspect is now emphasized in the Discussion, where we clarify that the present results do not question the nutritional value of pseudocereals, but rather underscore the importance of the form and proportion in which they are added to the formulation.

Comment 10: Conclusion: In the conclusion section, major improvements should be made, including the optimal formulation.

Response 10:  We appreciate your comment. In accordance with your suggestion, we have revised and strengthened the Conclusions section in order to summarize the main findings more clearly and explicitly identify the most appropriate formulation, considering the set of parameters evaluated.

Reviewer 3 Report

Comments and Suggestions for Authors

The manuscript presents a well-conceived and executed study on the development of cereal bars enriched with Andean grains and calafate. The research is highly relevant, and the experimental design effectively isolates the impact of key ingredients through four distinct formulations. The combination of instrumental and sensory analysis provides a comprehensive evaluation. The manuscript is suitable for publication after minor revisions addressing the following points:

  1. The crucial detail that the "Nutritional composition was estimated theoretically" is present but could be emphasized more prominently in the abstract to prevent any potential misinterpretation as analytically determined data from the outset. A slight rephrasing for clarity, such as "...was theoretically estimated using food composition tables...", is recommended.
  2. The section 2.4 title, "Determination of Nutritional Analyses," is slightly awkward, as "Analyses" refers to the process rather than the subject. A more precise and standard title would be "Theoretical Estimation of Nutritional Composition", which accurately describes the methodology used in this section.
  3. There is a verbatim repetition of the sentencein section 3.1: "Similar interactions between phenolic compounds and polysaccharides have been demonstrated to enhance the structural integrity and modify the textural properties of cereal-based food systems." One instance should be deleted to improve conciseness.
  4. Throughout the manuscript, the significance level is indicated with an uppercase 'P' (e.g., P < 0.05). The conventional notation in scientific literature is a lowercase, italicized *p* (i.e., *p* < 0.05). Please correct this for consistency with standard practice.
  5. In Line 530,the term "substantial products" is unclear in the context of texture. Based on the surrounding discussion, this likely refers to "excessively hard products" or products with "high hardness," which would more effectively convey the point that extreme firmness negatively impacts acceptability.
  6. In line 204,the phrase "effective tests" appears to be a minor error. Given the context of measuring consumer liking and preference, the correct terminology is "affective tests."
  7. The descriptions for the formulationsin figure 1, particularly F1 ("balanced proportions") and F4 ("Control cereal bar"), are somewhat vague. Directly referencing the key differentiating variables from Table 1 would enhance clarity (e.g., "F1: Baseline formulation," "F4: Oats-only control").
  8. The table is informative but complex. The formatting could be simplified for better readability. For instance, the column header "Servin g size" contains an erroneous space. Furthermore, the footnote "mg size" is ambiguous and likely intended to mean "per 100g".
  9. To further strengthen the discussion regarding the modest protein and fiber content, a brief speculative comment on the potential magnitude of improvement achievable by using whole grains would be valuable for guiding future research.

Author Response

Ref.: Manuscript ID: Foods- 3992770

Responses to Reviewers' comments

Summary

Thank you very much for handling our manuscript entitled “Development of cereal bars enriched with Andean grains and Patagonian calafate (Berberis microphylla): nutritional compo-sition, phenolics, antioxidant, textural, and sensory evaluation” (Manuscript ID: Foods-3992770). We greatly appreciate the opportunity that you have given us to review further this manuscript.

Thank you very much to the reviewers for the constructive comments. The authors have done their best to attend to all the comments, either by rewriting or amending the manuscript or justifying the action. Based on the valuable remarks given to us, the manuscript was redrafted in several sections. We trust the manuscript is more fluent and understandable now.

We are pleased that the referees found our study interesting, and we appreciate their comments, which have helped us to improve our manuscript.

Please find below our reply to these comments. All changes in the revised manuscript were highlighted by using red text. We hope that you will find our manuscript worthy of publication in Foods.

Reviewer #3:

The manuscript presents a well-conceived and executed study on the development of cereal bars enriched with Andean grains and calafate. The research is highly relevant, and the experimental design effectively isolates the impact of key ingredients through four distinct formulations. The combination of instrumental and sensory analysis provides a comprehensive evaluation. The manuscript is suitable for publication after minor revisions addressing the following points:

Comment 1: The crucial detail that the "Nutritional composition was estimated theoretically" is present but could be emphasized more prominently in the abstract to prevent any potential misinterpretation as analytically determined data from the outset. A slight rephrasing for clarity, such as "...was theoretically estimated using food composition tables...", is recommended.

Response 1:  We appreciate the reviewer's comment. The previous version of the abstract already indicated that the nutritional composition was “estimated theoretically from composition tables and DRIs.” However, to avoid any possible ambiguity, we have adjusted the wording following the reviewer's recommendation, so that the theoretical nature of these data is clear from the outset.

Comment 2: The section 2.4 title, "Determination of Nutritional Analyses," is slightly awkward, as "Analyses" refers to the process rather than the subject. A more precise and standard title would be "Theoretical Estimation of Nutritional Composition", which accurately describes the methodology used in this section.

Response 2: Dear reviewer, we appreciate your comment and fully accept it. We believe that this title more accurately describes the methodology used, so the title will be changed as you suggested: “Theoretical Estimation of Nutritional Composition.”

Comment 3: There is a verbatim repetition of the sentence in section 3.1: "Similar interactions between phenolic compounds and polysaccharides have been demonstrated to enhance the structural integrity and modify the textural properties of cereal-based food systems." One instance should be deleted to improve conciseness.

Response 3: Thank you for your comment. In the revised version of the manuscript, we have removed the repetition of the phrase in section 3.1, keeping only one instance to improve the conciseness of the text.

Comment 4: Throughout the manuscript, the significance level is indicated with an uppercase 'P' (e.g., P < 0.05). The conventional notation in scientific literature is a lowercase, italicized *p* (i.e., *p* < 0.05). Please correct this for consistency with standard practice.

Response 4: We appreciate the comment. In the revised version of the manuscript, all occurrences of “P” have been replaced with the conventional notation using lowercase italic p (e.g., p < 0.05), in accordance with standard practice in scientific literature.

Comment 5: In Line 530, the term "substantial products" is unclear in the context of texture. Based on the surrounding discussion, this likely refers to "excessively hard products" or products with "high hardness," which would more effectively convey the point that extreme firmness negatively impacts acceptability.

Response 5: We thank the reviewer for this constructive suggestion. In the revised version of the manuscript, we have replaced the ambiguous term “substantial products” with the expression “high hardness products” to more accurately reflect that extreme firmness negatively affects acceptability, as you suggest.

Comment 6: In line 204,the phrase "effective tests" appears to be a minor error. Given the context of measuring consumer liking and preference, the correct terminology is "affective tests."

Response 6: Thank you for the correction. In the revised version of the manuscript, we have replaced the term “effective tests” with the correct expression “affective tests.”

Comment 7: The descriptions for the formulations in figure 1, particularly F1 ("balanced proportions") and F4 ("Control cereal bar"), are somewhat vague. Directly referencing the key differentiating variables from Table 1 would enhance clarity (e.g., "F1: Baseline formulation," "F4: Oats-only control").

Response 7:  We appreciate your comment. In the revised version of the manuscript, we have modified the description of the formulations in Figure 1 to refer directly to the differentiating variables indicated in Table 1, replacing vague expressions such as “balanced proportions” and “control cereal bar.”

Comment 8: The table is informative but complex. The formatting could be simplified for better readability. For instance, the column header "Servin g size" contains an erroneous space. Furthermore, the footnote "mg size" is ambiguous and likely intended to mean "per 100g".

Response 8: Dear reviewer, Table 4: Nutritional components of different formulations of cereal bars plays a key comparative role within the manuscript, as it not only presents nutritional values but also allows for a direct comparison of the four formulations (F1–F4) in terms of macronutrients, micronutrients, and percentage of daily requirements. Furthermore, the inclusion of information per 100 g and per serving adds greater scientific value to the study, as it facilitates comparison with the literature (commonly reported per 100 g) and, at the same time, allows for interpretation from the consumer's point of view, considering an actual serving size.

On the other hand, because this study is based on a theoretical analysis of nutritional content, using food composition tables (INTA, USDA, and DRIs) rather than direct laboratory analytical determinations, the table plays a central role in demonstrating the methodological rigor employed. Excessive simplification of this could give the impression of a superficial analysis and limit a comprehensive understanding of the differences between the formulations.

Regarding your comment on “mg size,” we clarify that it actually corresponds to “serving size” and that, due to a formatting error in the table, the word was separated, causing confusion. This situation will be corrected in the revised version of the manuscript to ensure adequate clarity in the presentation of the table.

Comment 9: To further strengthen the discussion regarding the modest protein and fiber content, a brief speculative comment on the potential magnitude of improvement achievable by using whole grains would be valuable for guiding future research.

Response 9: Thank you for your excellent comment. Indeed, a promising way to increase the protein and fiber content in our formulations would be to replace “pop” (expanded) grains with whole or minimally processed versions. Although we did not conduct this experiment in the present study, recent studies suggest that the use of whole grains has a significant nutritional impact: for example, the incorporation of whole or sprouted pseudocereals (such as quinoa or amaranth) in food products has shown significant increases in total protein and dietary fiber. Therefore, future research should incorporate whole or sprouted grains into formulations in order to optimize their nutritional and functional profile. This was integrated into the text in lines 448-456.

Reviewer 4 Report

Comments and Suggestions for Authors

The manuscript aims to formulate bars enriched with quinoa, amaranth, and calafate (Berberis microphylla) and evaluate the bars' instrumental texture, total phenolic compounds, antioxidant capacity, nutritional composition, and sensory evaluation. The title does not fully reflect the study and should be improved. The introduction is generally sufficient to understand the background of the study but could be expanded upon. The objective is well-defined.  Overall, the manuscript presents important scientific findings based on tables and figures, but the comparison of these results with similar literature studies should be improved. The weaknesses of the study include the lack of optimization in the selection of ingredient amounts in the bar formulation and the assessment of nutritional composition using only standard tables. Furthermore, the major concern regarding this manuscript is the number of replications. According to the materials and methods, the authors did not report any number of replicates for bar production to determine variability. Just below I suggest some  changes that might improve the manuscript.

-Line 97: Please clearly and in detail explain how the current study differs from cereal bars previously produced with pseudocereals.

Line 120: Please include a reference for cereal bar production.

Line 120: Please specify how many replicates were carried out in bar production.

Line 121: Please detail the pretreatments applied to these ingredients in the method section.

Line 122: Blueberries or cranberries? Please confirm and use the same term throughout the manuscript.

Line 131: On what basis were the ingredient amounts selected for the formulations? Considering all the results, why was a limited sample size chosen rather than optimization for the production of cereal bars with high bioactive properties and improved sensory properties?

Line 189: ET or TE? Please confirm and use the same term throughout the manuscript.

Lines 282-284: Please express the sentence in a different way.

Line 285: Despite the decreased oat content and increased pseudocereal content in F2 compared to F1, TPC and antioxidant activity were negatively affected. Therefore, presenting the TPC and antioxidant activity properties of oat, amaranth popcorn, quinoa popcorn, and dehydrated berries is important for a better understanding of the results.

Line 364: Theoretical estimates of nutritional composition values ​​calculated from standard composition tables are valuable for enriching the manuscript. However, I recommend that these data be supported by laboratory assays and compared with similar cereal bars in the literature.

Lines 512-514: Since no optimization was applied to the formulations in the study, please avoid making assertive statements about the amounts that could achieve a more balanced matrix.

-Line 550: In the conclusion section, please also present the shortcomings of the current study and your suggestions for future studies.

Author Response

Ref.: Manuscript ID: Foods- 3992770

Responses to Reviewers' comments

Summary

Thank you very much for handling our manuscript entitled “Development of cereal bars enriched with Andean grains and Patagonian calafate (Berberis microphylla): nutritional compo-sition, phenolics, antioxidant, textural, and sensory evaluation” (Manuscript ID: Foods-3992770). We greatly appreciate the opportunity that you have given us to review further this manuscript.

Thank you very much to the reviewers for the constructive comments. The authors have done their best to attend to all the comments, either by rewriting or amending the manuscript or justifying the action. Based on the valuable remarks given to us, the manuscript was redrafted in several sections. We trust the manuscript is more fluent and understandable now.

We are pleased that the referees found our study interesting, and we appreciate their comments, which have helped us to improve our manuscript.

Please find below our reply to these comments. All changes in the revised manuscript were highlighted by using red text. We hope that you will find our manuscript worthy of publication in Foods.

Reviewer #4:

The manuscript aims to formulate bars enriched with quinoa, amaranth, and calafate (Berberis microphylla) and evaluate the bars' instrumental texture, total phenolic compounds, antioxidant capacity, nutritional composition, and sensory evaluation. The title does not fully reflect the study and should be improved. The introduction is generally sufficient to understand the background of the study but could be expanded upon. The objective is well-defined.  Overall, the manuscript presents important scientific findings based on tables and figures, but the comparison of these results with similar literature studies should be improved. The weaknesses of the study include the lack of optimization in the selection of ingredient amounts in the bar formulation and the assessment of nutritional composition using only standard tables. Furthermore, the major concern regarding this manuscript is the number of replications. According to the materials and methods, the authors did not report any number of replicates for bar production to determine variability. Just below I suggest some  changes that might improve the manuscript.

Comment 1: Line 97: Please clearly and in detail explain how the current study differs from cereal bars previously produced with pseudocereals.

Response 1: We appreciate your comment. To explain more clearly how our study differs from previous work on cereal bars made with pseudocereals, we would like to point out that previously reported studies on bars enriched with quinoa and/or amaranth have focused mainly on describing the nutritional composition and some technological parameters (such as texture or stability), generally using a single pseudocereal or incorporating it as a complementary ingredient. Therefore, what differentiates our study is: 1) Our study formulates cereal bars that simultaneously integrate two Andean pseudocereals (quinoa and amaranth) with a native berry with high functional potential (calafate, Berberis microphylla), which allows us to evaluate the combined contribution of these regional ingredients in the same matrix. 2) It comprehensively evaluates the theoretical nutritional composition, total phenolic compound content, antioxidant capacity, instrumental texture, and sensory acceptance, rather than limiting itself to only the proximate composition or technological quality. 3) Includes different proportions of pseudocereals and calafate in four formulations, allowing analysis of how increasing these functional ingredients simultaneously modulates the functional (phenols and antioxidant activity), mechanical (hardness), and sensory properties of these bars.

Comment 2: Line 120: Please include a reference for cereal bar production.

Response 2:  We appreciate the reviewer's suggestion. The procedure for making the cereal bars used in this study was based on the stages traditionally used for this type of product (mixing dry and wet ingredients, compacting in a tray, baking, cooling, and cutting), adjusted through our own preliminary tests, rather than on a specific protocol taken from the literature. Therefore, the manuscript provides a detailed description of the manufacturing process to ensure its complete reproducibility, detailing the stages of mixing, forming, baking, cooling, and cutting, as well as the process conditions (times and temperatures). Since no single published protocol was used as the primary source, we consider it more appropriate to clearly describe the procedure developed in this work, rather than attributing it to a specific reference that was not actually used as a guide.

Comment 3: Line 120: Please specify how many replicates were carried out in bar production.

Response 3: We appreciate the reviewer's comment. In this study, for each formulation, three independent batches of cereal bars were produced, following the same production procedure. The samples used in the various analyses were obtained from these batches. This information was added to the manuscript in the materials and methods section.

Comment 4: Line 121: Please detail the pretreatments applied to these ingredients in the method section.

Response 4: We appreciate your comment. In this study, the dry ingredients used (oats, expanded amaranth, expanded quinoa, dried cranberries, and powdered calafate) were purchased from commercial suppliers in their ready-to-use form, so no additional pretreatments were performed in the laboratory. In the revised version of the manuscript, we have clarified this point in the Materials and Methods section, explicitly stating that the ingredients were used as purchased and were not pretreated by us.

Comment 5: Line 122: Blueberries or cranberries? Please confirm and use the same term throughout the manuscript.

Response 5: We appreciate your observation. The correct ingredient used in the formulations is dried cranberries, not blueberries. In the revised version of the manuscript, we have corrected this, replacing “blueberries” with “cranberries.”

Comment 6: Line 131: On what basis were the ingredient amounts selected for the formulations? Considering all the results, why was a limited sample size chosen rather than optimization for the production of cereal bars with high bioactive properties and improved sensory properties?

Response 6:  Thank you very much for your comment. First of all, the quantities of ingredients used in the four formulations were defined based on a combination of technological, nutritional, and sensory criteria, supported by preliminary tests: 1) A working range was established in which the mixture maintained good cohesion, stable structure after baking, and ease of removal from the mold and cutting, which limited both the maximum level of expanded pseudocereals and powdered calafate. 2) The amount of honey and oil was adjusted to ensure adequate binding and a chewable texture, avoiding excessively sticky or fragile formulations, while maintaining a reasonable nutritional profile in terms of fats and sugars. 3) As for the amount of calafate, it was added to the point where the dough remained workable and the intensity of flavor, acidity, and astringency was sensorially acceptable in preliminary trials; higher levels negatively affected the sensory profile. 4) The cranberry content was kept constant in all formulations so as not to introduce a second factor of variation in the berry content, so that the changes observed could be attributed mainly to variations in pseudocereals and calafate. Thus, the four formulations represent a deliberate gradient in the inclusion of Andean pseudocereals and calafate, within a technologically viable range compatible with a realistic consumer produRegarding the question of optimization, the objective of this work was set out as a study of the development and comprehensive characterization of cereal bars enriched with regional ingredients, rather than as an optimization study. The implementation of an optimization design (e.g., a factorial or response surface design with various levels of calafate, pseudocereals, and oats) would have required a much larger number of formulations, which exceeds the scope and resources of this study. Therefore, each formulation was evaluated in depth (theoretical nutritional composition, phenolic compounds, antioxidant capacity, instrumental texture, and sensory acceptance), which involved considerable use of samples, laboratory time, and consumer participation.

However, we recognize that fine-tuning the levels of each ingredient is an important step, and we have therefore explicitly included it as a projection for future work in the conclusion.

Comment 7: Line 189: ET or TE? Please confirm and use the same term throughout the manuscript.

Response 7: Thank you for your comment. In line 189 and throughout the manuscript, we have corrected the abbreviation to TE (Trolox Equivalents) and standardized the nomenclature to ensure consistency throughout the manuscript.

Comment 8: Lines 282-284: Please express the sentence in a different way.

Response 8: We appreciate the reviewer's comment. We agree that the sentence could be expressed more clearly. In the revised version of the manuscript, we have reworded the sentence to better convey the idea.

Comment 9: Line 285: Despite the decreased oat content and increased pseudocereal content in F2 compared to F1, TPC and antioxidant activity were negatively affected. Therefore, presenting the TPC and antioxidant activity properties of oat, amaranth popcorn, quinoa popcorn, and dehydrated berries is important for a better understanding of the results.

Response 9: We appreciate the reviewer's comment and agree that knowing the phenolic content and antioxidant capacity of the individual ingredients is relevant for interpreting the results. However, in the present study, the total phenolic compound content and antioxidant capacity of each ingredient separately were not determined experimentally, but only for the bars already formulated. It is not possible to perform these analyses retrospectively, given that the original batches of raw materials and bars are no longer available, and this would involve a new experimental process that exceeds the scope and resources of this work.

In our study, formulation F1 had a higher total phenolic compound (TPC) content than F2, while both showed statistically similar antioxidant capacity as measured by DPPH, as they are in the same homogeneous group (see Table 3). This apparent discrepancy can be explained by several factors:

1) Although the difference in TPC between F1 and F2 is statistically significant, it is relatively moderate and falls within a concentration range in which the DPPH assay tends to reach a near-plateau response. Under these conditions, small increases in TPC do not necessarily translate into proportional increases in DPPH radical inhibition.

2) The Folin–Ciocalteu method quantifies the overall reducing capacity of the sample and responds to a broad spectrum of phenolic compounds as well as other non-phenolic reducing compounds. In contrast, the DPPH assay is more sensitive to those compounds capable of donating hydrogen or electrons particularly efficiently. Therefore, F1 could contain a greater amount of phenols that react with the Folin–Ciocalteu reagent but contribute less to DPPH radical scavenging compared to the phenolic profile of sample F2.

3) The differences in the proportion of oats, expanded pseudocereals, and calafate between F1 and F2 modify the structure of the food matrix.

These modifications can influence the release, solubility, and accessibility of compounds in the extraction solvent , thereby affecting their interaction with the DPPH radical, and resulting in similar antioxidant capacity values despite differences in TPC.

However, when all formulations (F1–F4) are evaluated together, a consistent trend between TPC and antioxidant capacity is maintained, as formulation F3 (with the highest level of calafate) has the highest values for both determinations, while F4 (without calafate or pseudocereals) has the lowest values.

Comment 10: Line 364: Theoretical estimates of nutritional composition values ​​calculated from standard composition tables are valuable for enriching the manuscript. However, I recommend that these data be supported by laboratory assays and compared with similar cereal bars in the literature.

Response 10:   We appreciate the reviewer's valuable suggestion and agree that conducting experimental nutritional analyses would provide very relevant information and allow us to directly compare our results with similar cereal bars reported in the literature. However, in the original design of this study, the main focus was on the incorporation of calafate and Andean pseudocereals and their effect on the phenolic compound content, antioxidant capacity, instrumental texture, and sensory acceptance of the bars. In this context, nutritional information was used as a descriptive complement and was estimated from the formulation and official food composition tables, without being considered a primary response variable.

Performing complete proximate analyses in the laboratory (moisture, protein, lipids, ash, fiber, and carbohydrates by difference) for the four formulations, with their respective replicates, involves considerable consumption of samples, reagents, and instrumental time, which was not contemplated in the experimental plan or in the project resources. Including this battery of analyses now exceeds the scope of the study and is not feasible within the deadlines set for the manuscript review.

However, we acknowledge this limitation and have made it more explicit in the manuscript. The Materials and Methods section specifies that the reported nutritional composition is based on theoretical values calculated from food composition tables, and the Discussion section notes that these data should be considered approximations and recommends that future research include experimental proximate analyses of the bars and direct comparisons with similar products available in the literature.

Comment 11: Lines 512-514: Since no optimization was applied to the formulations in the study, please avoid making assertive statements about the amounts that could achieve a more balanced matrix.

Response 11:  We appreciate your comment. In the revised version of the manuscript, we have reworded this sentence to indicate more cautiously that the better performance of F2.

Comment 12: Line 550: In the conclusion section, please also present the shortcomings of the current study and your suggestions for future studies.

Response 12:  We appreciate your comment. The conclusions section has been revised to explicitly include the main limitations of the study and to propose future lines of research.

Round 2

Reviewer 1 Report

Comments and Suggestions for Authors

I am satisfied with the current revision. No more concerns.

Reviewer 2 Report

Comments and Suggestions for Authors

Dear Authors,

After carefully reviewing the manuscript, I confirm that all the suggestions have been addressed.

The manuscript is now complete and significantly improved. It aligns with the objectives and author guidelines of the MDPI Foods Journal, and I believe it can be accepted for publication.

....

Reviewer 4 Report

Comments and Suggestions for Authors

Most of my comments have been addressed and revised by the authors.

Because the nutritional composition and phenolic profile were not experimentally determined in the study, the title does not fully reflect the study. Therefore, "Phenolic Content" would be more appropriate than "Phenolics" in the title.